# Altitude Training:
# Strong Bounds for Single-Layer Dropout

**Stefan Wager**\*, **William Fithian**\*, **Sida Wang**†, and **Percy Liang**\*,†
Departments of Statistics\* and Computer Science†
Stanford University, Stanford, CA-94305, USA
{swager, wfithian}@stanford.edu, {sidaw, pliang}@cs.stanford.edu

## Abstract

Dropout training, originally designed for deep neural networks, has been success-
ful on high-dimensional single-layer natural language tasks. This paper proposes
a theoretical explanation for this phenomenon: we show that, under a generative
Poisson topic model with long documents, dropout training improves the exponent
in the generalization bound for empirical risk minimization. Dropout achieves this
gain much like a marathon runner who practices at altitude: once a classifier learns
to perform reasonably well on training examples that have been artificially cor-
rupted by dropout, it will do very well on the uncorrupted test set. We also show
that, under similar conditions, dropout preserves the Bayes decision boundary and
should therefore induce minimal bias in high dimensions.

## 1  Introduction

Dropout training [1] is an increasingly popular method for regularizing learning algorithms. Dropout
is most commonly used for regularizing deep neural networks [2, 3, 4, 5], but it has also been found
to improve the performance of logistic regression and other single-layer models for natural language
tasks such as document classification and named entity recognition [6, 7, 8]. For single-layer linear
models, learning with dropout is equivalent to using "blankout noise" [9].

The goal of this paper is to gain a better theoretical understanding of why dropout regularization
works well for natural language tasks. We focus on the task of document classification using linear
classifiers where data comes from a generative Poisson topic model. In this setting, dropout effec-
tively deletes random words from a document during training; this corruption makes the training
examples harder. A classifier that *is* able to fit the training data will therefore receive an accuracy
boost at test time on the much easier uncorrupted examples. An apt analogy is altitude training,
where athletes practice in more difficult situations than they compete in. Importantly, our analysis
does not rely on dropout merely creating *more* pseudo-examples for training, but rather on dropout
creating *more challenging* training examples. Somewhat paradoxically, we show that removing in-
formation from training examples can induce a classifier that performs better at test time.

**Main Result**  Consider training the zero-one loss empirical risk minimizer (ERM) using dropout,
where each word is independently removed with probability $\delta \in (0, 1)$. For a class of Poisson
generative topic models, we show that dropout gives rise to what we call the *altitude training phe-
nomenon*: dropout improves the excess risk of the ERM by multiplying the exponent in its decay
rate by $1/(1 - \delta)$. This improvement comes at the cost of an additive term of $O(1/\sqrt{\lambda})$, where $\lambda$
is the average number of words per document. More formally, let $h^*$ and $\hat{h}_0$ be the expected and

---

S. Wager and W. Fithian are supported by a B.C. and E.J. Eaves Stanford Graduate Fellowship and NSF
VIGRE grant DMS–0502385 respectively.

empirical risk minimizers, respectively; let $h_\delta^*$ and $\hat{h}_\delta$ be the corresponding quantities for dropout training. Let $\mathrm{Err}(h)$ denote the error rate (on test examples) of $h$. In Section 4, we show that:

$$\underbrace{\mathrm{Err}\left(\hat{h}_\delta\right) - \mathrm{Err}\left(h_\delta^*\right)}_{\text{dropout excess risk}} = \widetilde{\mathcal{O}}_P\left(\underbrace{\left(\mathrm{Err}\left(\hat{h}_0\right) - \mathrm{Err}\left(h^*\right)\right)^{\frac{1}{1-\delta}}}_{\text{ERM excess risk}} + \frac{1}{\sqrt{\lambda}}\right), \tag{1}$$

where $\widetilde{\mathcal{O}}_P$ is a variant of big-$\mathcal{O}$ in probability notation that suppresses logarithmic factors. If $\lambda$ is large (we are classifying long documents rather than short snippets of text), dropout considerably accelerates the decay rate of excess risk. The bound (1) holds for fixed choices of $\delta$. The constants in the bound worsen as $\delta$ approaches 1, and so we cannot get zero excess risk by sending $\delta$ to 1.

Our result is modular in that it converts upper bounds on the ERM excess risk to upper bounds on the dropout excess risk. For example, recall from classic VC theory that the ERM excess risk is $\widetilde{\mathcal{O}}_P(\sqrt{d/n})$, where $d$ is the number of features (vocabulary size) and $n$ is the number of training examples. With dropout $\delta = 0.5$, our result (1) directly implies that the dropout excess risk is $\widetilde{\mathcal{O}}_P(d/n + 1/\sqrt{\lambda})$.

The intuition behind the proof of (1) is as follows: when $\delta = 0.5$, we essentially train on half documents and test on whole documents. By conditional independence properties of the generative topic model, the classification score is roughly Gaussian under a Berry-Esseen bound, and the error rate is governed by the tails of the Gaussian. Compared to half documents, the coefficient of variation of the classification score on whole documents (at test time) is scaled down by $\sqrt{1-\delta}$ compared to half documents (at training time), resulting in an exponential reduction in error. The additive penalty of $1/\sqrt{\lambda}$ stems from the Berry-Esseen approximation.

Note that the bound (1) only controls the dropout excess risk. Even if dropout reduces the excess risk, it may introduce a bias $\mathrm{Err}(h_\delta^*) - \mathrm{Err}(h^*)$, and thus (1) is useful only when this bias is small. In Section 5, we will show that the optimal Bayes decision boundary is not affected by dropout under the Poisson topic model. Bias is thus negligible when the Bayes boundary is close to linear.

It is instructive to compare our generalization bound to that of Ng and Jordan [10], who showed that the naive Bayes classifier exploits a strong generative assumption—conditional independence of the features given the label—to achieve an excess risk of $\mathcal{O}_P(\sqrt{(\log d)/n})$. However, if the generative assumption is incorrect, then naive Bayes can have a large bias. Dropout enables us to cut excess risk without incurring as much bias. In fact, naive Bayes is closely related to logistic regression trained using an extreme form of dropout with $\delta \to 1$. Training logistic regression with dropout rates from the range $\delta \in (0, 1)$ thus gives a family of classifiers between unregularized logistic regression and naive Bayes, allowing us to tune the bias-variance tradeoff.

**Other perspectives on dropout** In the general setting, dropout only improves generalization by a *multiplicative* factor. McAllester [11] used the PAC-Bayes framework to prove a generalization bound for dropout that decays as $1 - \delta$. Moreover, provided that $\delta$ is not too close to 1, dropout behaves similarly to an adaptive $L_2$ regularizer with parameter $\delta/(1-\delta)$ [6, 12], and at least in linear regression such $L_2$ regularization improves generalization error by a constant factor. In contrast, by leveraging the conditional independence assumptions of the topic model, we are able to improve the *exponent* in the rate of convergence of the empirical risk minimizer.

It is also possible to analyze dropout as an adaptive regularizer [6, 9, 13]: in comparison with $L_2$ regularization, dropout favors the use of rare features and encourages confident predictions. If we believe that good document classification should produce confident predictions by understanding rare words with Poisson-like occurrence patterns, then the work on dropout as adaptive regularization and our generalization-based analysis are two complementary explanations for the success of dropout in natural language tasks.

## 2   Dropout Training for Topic Models

In this section, we introduce *binomial dropout*, a form of dropout suitable for topic models, and the Poisson topic model, on which all our analyses will be based.

**Binomial Dropout** Suppose that we have a binary classification problem[1] with count features $x^{(i)} \in \{0, 1, 2, \ldots\}^d$ and labels $y^{(i)} \in \{0, 1\}$. For example, $x_j^{(i)}$ is the number of times the $j$-th word in our dictionary appears in the $i$-th document, and $y^{(i)}$ is the label of the document. Our goal is to train a weight vector $\widehat{w}$ that classifies new examples with features $x$ via a linear decision rule $\hat{y} = \mathbb{I}\{\widehat{w} \cdot x > 0\}$. We start with the usual empirical risk minimizer:

$$\widehat{w}_0 \stackrel{\text{def}}{=} \operatorname{argmin}_{w \in \mathbb{R}^d} \left\{ \sum_{i=1}^{n} \ell\left(w; x^{(i)}, y^{(i)}\right) \right\} \tag{2}$$

for some loss function $\ell$ (we will analyze the zero-one loss but use logistic loss in experiments [e.g., 10, 14, 15]). Binomial dropout trains on perturbed features $\tilde{x}^{(i)}$ instead of the original features $x^{(i)}$:

$$\widehat{w}_\delta \stackrel{\text{def}}{=} \operatorname{argmin}_w \left\{ \sum_{i=1}^{n} \mathbb{E}\left[ \ell\left(w; \tilde{x}^{(i)}, y^{(i)}\right) \right] \right\}, \text{ where } \tilde{x}_j^{(i)} = \operatorname{Binom}\left(x_j^{(i)}; 1 - \delta\right). \tag{3}$$

In other words, during training, we randomly thin the $j$-th feature $x_j$ with binomial noise. If $x_j$ counts the number of times the $j$-th word appears in the document, then replacing $x_j$ with $\tilde{x}_j$ is equivalent to independently deleting each occurrence of word $j$ with probability $\delta$. Because we are only interested in the decision boundary, we do not scale down the weight vector obtained by dropout by a factor $1 - \delta$ as is often done [e.g., 1].

Binomial dropout differs slightly from the usual definition of (blankout) dropout, which alters the feature vector $x$ by setting random coordinates to 0 [6, 9, 11, 12]. The reason we chose to study binomial rather than blankout dropout is that Poisson random variables remain Poisson even after binomial thinning; this fact lets us streamline our analysis. For rare words that appear once in the document, the two types of dropout are equivalent.

**A Generative Poisson Topic Model** Throughout our analysis, we assume that the data is drawn from a Poisson topic model depicted in Figure 1a and defined as follows. Each document $i$ is assigned a label $y^{(i)}$ according to some Bernoulli distribution. Then, given the label $y^{(i)}$, the document gets a topic $\tau^{(i)} \in \Theta$ from a distribution $\rho_{y^{(i)}}$. Given the topic $\tau^{(i)}$, for every word $j$ in the vocabulary, we generate its frequency $x_j^{(i)}$ according to $x_j^{(i)} \mid \tau^{(i)} \sim \operatorname{Poisson}(\lambda_j^{(\tau^{(i)})})$, where $\lambda_j^{(\tau)} \in [0, \infty)$ is the expected number of times word $j$ appears under topic $\tau$. Note that $\|\lambda^{(\tau)}\|_1$ is the average length of a document with topic $\tau$. Define $\lambda \stackrel{\text{def}}{=} \min_{\tau \in \Theta} \|\lambda^{(\tau)}\|_1$ to be the shortest average document length across topics. If $\Theta$ contains only two topics—one for each class—we get the naive Bayes model. If $\Theta$ is the $(K-1)$-dimensional simplex where $\lambda^{(\tau)}$ is a $\tau$-mixture over $K$ basis vectors, we get the $K$-topic latent Dirichlet allocation [16].[2]

Note that although our generalization result relies on a generative model, the actual learning algorithm is agnostic to it. Our analysis shows that dropout can take advantage of a generative structure while remaining a discriminative procedure. If we believed that a certain topic model held exactly and we knew the number of topics, we could try to fit the full generative model by EM. This, however, could make us vulnerable to model misspecification. In contrast, dropout benefits from generative assumptions while remaining more robust to misspecification.

## 3   Altitude Training: Linking the Dropout and Data-Generating Measures

Our goal is to understand the behavior of a classifier $\hat{h}_\delta$ trained using dropout. During dropout, the error of any classifier $h$ is characterized by two measures. In the end, we are interested in the usual generalization error (expected risk) of $h$ where $x$ is drawn from the underlying *data-generating measure*:

$$\operatorname{Err}(h) \stackrel{\text{def}}{=} \mathbb{P}[y \neq h(x)]. \tag{4}$$

However, since dropout training works on the corrupted data $\tilde{x}$ (see (3)), in the limit of infinite data, the dropout estimator will converge to the minimizer of the generalization error with respect to the *dropout measure* over $\tilde{x}$:

$$\mathrm{Err}_\delta\left(h\right) \overset{\text{def}}{=} \mathbb{P}\left[y \neq h(\tilde{x})\right]. \tag{5}$$

The main difficulty in analyzing the generalization of dropout is that classical theory tells us that the generalization error with respect to the dropout measure will decrease as $n \to \infty$, but we are interested in the original measure. Thus, we need to bound $\mathrm{Err}$ in terms of $\mathrm{Err}_\delta$. In this section, we show that the error on the original measure is actually much smaller than the error on the dropout measure; we call this the *altitude training phenomenon*.

Under our generative model, the count features $x_j$ are conditionally independent given the topic $\tau$. We thus focus on a single fixed topic $\tau$ and establish the following theorem, which provides a per-topic analogue of (1). Section 4 will then use this theorem to obtain our main result.

**Theorem 1.** *Let $h$ be a binary linear classifier with weights $w$, and suppose that our features are drawn from the Poisson generative model given topic $\tau$. Let $c_\tau$ be the more likely label given $\tau$:*

$$c_\tau \overset{\text{def}}{=} \arg \max_{c \in \{0,1\}} \mathbb{P}\left[y^{(i)} = c \,\big|\, \tau^{(i)} = \tau\right]. \tag{6}$$

*Let $\tilde{\varepsilon}_\tau$ be the sub-optimal prediction rate in the dropout measure*

$$\tilde{\varepsilon}_\tau \overset{\text{def}}{=} \mathbb{P}\left[\mathbb{I}\left\{w \cdot \tilde{x}^{(i)} > 0\right\} \neq c_\tau \,\big|\, \tau^{(i)} = \tau\right], \tag{7}$$

*where $\tilde{x}^{(i)}$ is an example thinned by binomial dropout (3), and $\mathbb{P}$ is taken over the data-generating process. Let $\varepsilon_\tau$ be the sub-optimal prediction rate in the original measure*

$$\varepsilon_\tau \overset{\text{def}}{=} \mathbb{P}\left[\mathbb{I}\left\{w \cdot x^{(i)} > 0\right\} \neq c_\tau \,\big|\, \tau^{(i)} = \tau\right]. \tag{8}$$

*Then:*

$$\varepsilon_\tau = \widetilde{\mathcal{O}}\left(\tilde{\varepsilon}_\tau^{\frac{1}{1-\delta}} + \sqrt{\Psi_\tau}\right), \tag{9}$$

*where $\Psi_\tau = \max_j \left\{w_j^2\right\}/\sum_{j=1}^d \lambda_j^{(\tau)} w_j^2$, and the constants in the bound depend only on $\delta$.*

Theorem 1 only provides us with a useful bound when the term $\Psi_\tau$ is small. Whenever the largest $w_j^2$ is not much larger than the average $w_j^2$, then $\sqrt{\Psi_\tau}$ scales as $O(1/\sqrt{\lambda})$, where $\lambda$ is the average document length. Thus, the bound (9) is most useful for long documents.

**A Heuristic Proof of Theorem 1.** The proof of Theorem 1 is provided in the technical appendix. Here, we provide a heuristic argument for intuition. Given a fixed topic $\tau$, suppose that it is optimal to predict $c_\tau = 1$, so our test error is $\varepsilon_\tau = \mathbb{P}\left[w \cdot x \leq 0 \,\big|\, \tau\right]$. For long enough documents, by the central limit theorem, the score $s \overset{\text{def}}{=} w \cdot x$ will be roughly Gaussian $s \sim \mathcal{N}\left(\mu_\tau, \sigma_\tau^2\right)$, where $\mu_\tau = \sum_{j=1}^d \lambda_j^{(\tau)} w_j$ and $\sigma_\tau^2 = \sum_{j=1}^d \lambda_j^{(\tau)} w_j^2$. This implies that $\varepsilon_\tau \approx \Phi\left(-\mu_\tau/\sigma_\tau\right)$, where $\Phi$ is the cumulative distribution function of the Gaussian. Now, let $\tilde{s} \overset{\text{def}}{=} w \cdot \tilde{x}$ be the score on a dropout sample. Clearly, $\mathbb{E}\left[\tilde{s}\right] = \left(1 - \delta\right)\mu_\tau$ and $\mathrm{Var}\left[\tilde{s}\right] = \left(1 - \delta\right)\sigma_\tau^2$, because the variance of a Poisson random variable scales with its mean. Thus,

$$\tilde{\varepsilon}_\tau \approx \Phi\left(-\sqrt{1-\delta}\,\frac{\mu_\tau}{\sigma_\tau}\right) \approx \Phi\left(-\frac{\mu_\tau}{\sigma_\tau}\right)^{(1-\delta)} \approx \varepsilon_\tau^{(1-\delta)}. \tag{10}$$

Figure 1b illustrates the relationship between the two Gaussians. This explains the first term on the right-hand side of (9). The extra error term $\sqrt{\Psi_\tau}$ arises from a Berry-Esseen bound that approximates Poisson mixtures by Gaussian random variables.

## 4 A Generalization Bound for Dropout

By setting up a bridge between the dropout measure and the original data-generating measure, Theorem 1 provides a foundation for our analysis. It remains to translate this result into a statement about the generalization error of dropout. For this, we need to make a few assumptions.

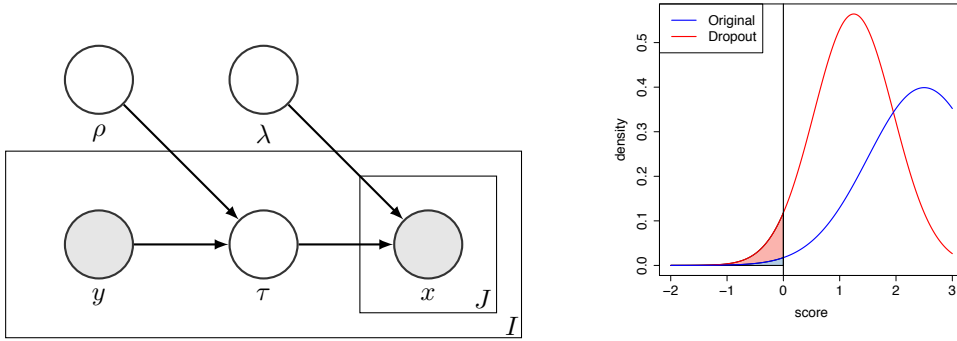

(a) Graphical representation of the Poisson topic model: Given a document with label $y$, we draw a document topic $\tau$ from the multinomial distribution with probabilities $\rho_y$. Then, we draw the words $x$ from the topic's Poisson distribution with mean $\lambda^{(\tau)}$. Boxes indicate repeated observations, and greyed-out nodes are observed during training.

(b) For a fixed classifier $w$, the probabilities of error on an example drawn from the original and dropout measures are governed by the tails of two Gaussians (shaded). The dropout Gaussian has a larger coefficient of variation, which means the error on the original measure (test) is much smaller than the error on the dropout measure (train) (10). In this example, $\mu = 2.5$, $\sigma = 1$, and $\delta = 0.5$.

Figure 1: (a) Graphical model. (b) The altitude training phenomenon.

Our first assumption is fundamental: if the classification signal is concentrated among just a few features, then we cannot expect dropout training to do well. The second and third assumptions, which are more technical, guarantee that a classifier can only do well overall if it does well on every topic; this lets us apply Theorem 1. A more general analysis that relaxes Assumptions 2 and 3 may be an interesting avenue for future work.

**Assumption 1: well-balanced weights**  First, we need to assume that all the signal is not concentrated in a few features. To make this intuition formal, we say a linear classifier with weights $w$ is *well-balanced* if the following holds for each topic $\tau$:

$$\frac{\max_j \left\{w_j^2\right\} \sum_{j=1}^d \lambda_j^{(\tau)}}{\sum_{j=1}^d \lambda_j^{(\tau)} w_j^2} \leq \kappa \text{ for some } 0 < \kappa < \infty. \tag{11}$$

For example, suppose each word was either useful ($|w_j| = 1$) or not ($w_j = 0$); then $\kappa$ is the inverse expected fraction of words in a document that are useful. In Theorem 2 we restrict the ERM to well-balanced classifiers and assume that the expected risk minimizer $h^*$ over all linear rules is also well-balanced.

**Assumption 2: discrete topics**  Second, we assume that there are a finite number $T$ of topics, and that the available topics are not too rare or ambiguous: the minimal probability of observing any topic $\tau$ is bounded below by

$$\mathbb{P}\left[\tau\right] \geq p_{\min} > 0, \tag{12}$$

and that each topic-conditional probability is bounded away from $\frac{1}{2}$ (random guessing):

$$\left| \mathbb{P}\left[y^{(i)} = c \,\middle|\, \tau^{(i)} = \tau\right] - \frac{1}{2} \right| \geq \alpha > 0 \tag{13}$$

for all topics $\tau \in \{1, ..., T\}$. This assumption substantially simplifies our arguments, allowing us to apply Theorem 1 to each topic separately without technical overhead.

**Assumption 3: distinct topics**  Finally, as an extension of Assumption 2, we require that the topics be "well separated." First, define $\text{Err}_{\min} = \mathbb{P}[y^{(i)} \neq c_{\tau^{(i)}}]$, where $c_\tau$ is the most likely label given topic $\tau$ (6); this is the error rate of the optimal decision rule that sees topic $\tau$. We assume that the best linear rule $h_\delta^*$ satisfying (11) is almost as good as always guessing the best label $c_\tau$ under the dropout measure:

$$\text{Err}_\delta \left(h_\delta^*\right) = \text{Err}_{\min} + \mathcal{O}\left(\frac{1}{\sqrt{\lambda}}\right), \tag{14}$$

where, as usual, $\lambda$ is a lower bound on the average document length. If the dimension $d$ is larger than the number of topics $T$, this assumption is fairly weak: the condition (14) holds whenever the matrix $\Pi$ of topic centers has full rank, and the minimum singular value of $\Pi$ is not too small (see Proposition 6 in the Appendix for details). This assumption is satisfied if the different topics can be separated from each other with a large margin.

Under Assumptions 1–3 we can turn Theorem 1 into a statement about generalization error.

**Theorem 2.** *Suppose that our features $x$ are drawn from the Poisson generative model (Figure 1a), and Assumptions 1–3 hold. Define the excess risks of the dropout classifier $\hat{h}_\delta$ on the dropout and data-generating measures, respectively:*

$$\tilde{\eta} \stackrel{\text{def}}{=} \mathrm{Err}_\delta\left(\hat{h}_\delta\right) - \mathrm{Err}_\delta\left(h_\delta^*\right) \quad and \quad \eta \stackrel{\text{def}}{=} \mathrm{Err}\left(\hat{h}_\delta\right) - \mathrm{Err}\left(h_\delta^*\right). \tag{15}$$

*Then, the altitude training phenomenon applies:*

$$\eta = \widetilde{\mathcal{O}}\left(\tilde{\eta}^{\frac{1}{1-\delta}} + \frac{1}{\sqrt{\lambda}}\right). \tag{16}$$

*The above bound scales linearly in $p_{min}^{-1}$ and $\alpha^{-1}$; the full dependence on $\delta$ is shown in the appendix.*

In a sense, Theorem 2 is a meta-generalization bound that allows us to transform generalization bounds with respect to the dropout measure ($\tilde{\eta}$) into ones on the data-generating measure ($\eta$) in a modular way. As a simple example, standard VC theory provides an $\tilde{\eta} = \widetilde{\mathcal{O}}_P(\sqrt{d/n})$ bound which, together with Theorem 2, yields:

**Corollary 3.** *Under the same conditions as Theorem 2, the dropout classifier $\hat{h}_\delta$ achieves the following excess risk:*

$$\mathrm{Err}\left(\hat{h}_\delta\right) - \mathrm{Err}\left(h_\delta^*\right) = \widetilde{\mathcal{O}}_P\left(\left(\sqrt{\frac{d}{n}}\right)^{\frac{1}{1-\delta}} + \frac{1}{\sqrt{\lambda}}\right). \tag{17}$$

More generally, we can often check that upper bounds for $\mathrm{Err}(\hat{h}) - \mathrm{Err}(h^*)$ also work as upper bounds for $\mathrm{Err}_\delta(\hat{h}_\delta) - \mathrm{Err}_\delta(h_\delta^*)$; this gives us the heuristic result from (1).

## 5   The Bias of Dropout

In the previous section, we showed that under the Poisson topic model in Figure 1a, dropout can achieve a substantial cut in excess risk $\mathrm{Err}(\hat{h}_\delta) - \mathrm{Err}(h_\delta^*)$. But to complete our picture of dropout's performance, we must address the bias of dropout: $\mathrm{Err}(h_\delta^*) - \mathrm{Err}(h^*)$.

Dropout can be viewed as importing "hints" from a generative assumption about the data. Each observed $(x, y)$ pair (each labeled document) gives us information not only about the conditional class probability at $x$, but also about the conditional class probabilities at numerous other hypothetical values $\tilde{x}$ representing shorter documents of the same class that did not occur. Intuitively, if these $\tilde{x}$ are actually good representatives of that class, the bias of dropout should be mild.

For our key result in this section, we will take the Poisson generative model from Figure 1a, but further assume that document length is independent of the topic. Under this assumption, we will show that dropout preserves the Bayes decision boundary in the following sense:

**Proposition 4.** *Let $(x, y)$ be distributed according to the Poisson topic model of Figure 1a. Assume that document length is independent of topic: $\|\lambda^{(\tau)}\|_1 = \lambda$ for all topics $\tau$. Let $\tilde{x}$ be a binomial dropout sample of $x$ with some dropout probability $\delta \in (0, 1)$. Then, for every feature vector $v \in \mathbb{R}^d$, we have:*

$$\mathbb{P}\left[y = 1 \,\middle|\, \tilde{x} = v\right] = \mathbb{P}\left[y = 1 \,\middle|\, x = v\right]. \tag{18}$$

If we had an infinite amount of data $(\tilde{x}, y)$ corrupted under dropout, we would predict according to $\mathbb{I}\{\mathbb{P}\left[y = 1 \,\middle|\, \tilde{x} = v\right] > \frac{1}{2}\}$. The significance of Proposition 4 is that this decision rule is identical to the true Bayes decision boundary (without dropout). Therefore, the empirical risk minimizer of a sufficiently rich hypothesis class trained with dropout would incur very small bias.

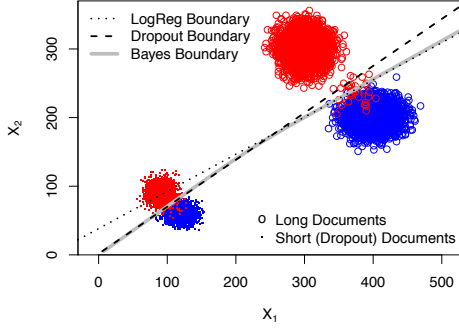

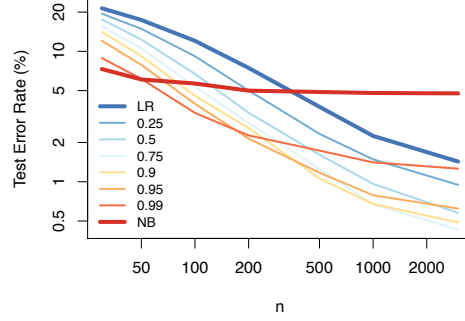

(a) Dropout ($\delta = 0.75$) with $d = 2$. For long documents (circles in the upper-right), logistic regression focuses on capturing the small red cluster; the large red cluster has almost no influence. Dropout (dots in the lower-left) distributes influence more equally between the two red clusters.

(b) Learning curves for the synthetic experiment. Each axis is plotted on a log scale. Here the dropout rate $\delta$ ranges from 0 (logistic regression) to 1 (naive Bayes) for multiple values of training set sizes $n$. As $n$ increases, less dropout is preferable, as the bias-variance tradeoff shifts.

Figure 2: Behavior of binomial dropout in simulations. In the left panel, the circles are the original data, while the dots are dropout-thinned examples. The Monte Carlo error is negligible.

However, Proposition 4 does *not* guarantee that dropout incurs no bias when we fit a linear classifier. In general, the best linear approximation for classifying shorter documents is not necessarily the best for classifying longer documents. As $n \to \infty$, a linear classifier trained on $(x, y)$ pairs will eventually outperform one trained on $(\tilde{x}, y)$ pairs.

**Dropout for Logistic Regression**    To gain some more intuition about how dropout affects linear classifiers, we consider logistic regression. A similar phenomenon should also hold for the ERM, but discussing this solution is more difficult since the ERM solution does not have have a simple characterization. The relationship between the 0-1 loss and convex surrogates has been studied by, e.g., [14, 15]. The score criterion for logistic regression is $0 = \sum_{i=1}^{n} \left( y^{(i)} - \hat{p}_i \right) x^{(i)}$, where $\hat{p}_i = (1 + e^{-\hat{w} \cdot x^{(i)}})^{-1}$ are the fitted probabilities. Note that easily-classified examples (where $\hat{p}_i$ is close to $y^{(i)}$) play almost no role in driving the fit. Dropout turns easy examples into hard examples, giving more examples a chance to participate in learning a good classification rule.

Figure 2a illustrates dropout's tendency to spread influence more democratically for a simple classification problem with $d = 2$. The red class is a 99:1 mixture over two topics, one of which is much less common, but harder to classify, than the other. There is only one topic for the blue class. For long documents (open circles in the top right), the infrequent, hard-to-classify red cluster dominates the fit while the frequent, easy-to-classify red cluster is essentially ignored. For dropout documents with $\delta = 0.75$ (small dots, lower left), both red clusters are relatively hard to classify, so the infrequent one plays a less disproportionate role in driving the fit. As a result, the fit based on dropout is more stable but misses the finer structure near the decision boundary. Note that the solid gray curve, the Bayes boundary, is unaffected by dropout, per Proposition 4. But, because it is nonlinear, we obtain a different linear approximation under dropout.

## 6    Experiments and Discussion

**Synthetic Experiment**    Consider the following instance of the Poisson topic model: We choose the document label uniformly at random: $\mathbb{P}\left[ y^{(i)} = 1 \right] = \frac{1}{2}$. Given label 0, we choose topic $\tau^{(i)} = 0$ deterministically; given label 1, we choose a real-valued topic $\tau^{(i)} \sim \mathrm{Exp}(3)$. The per-topic Poisson intensities $\lambda^{(\tau)}$ are defined as follows:

$$\theta^{(\tau)} = \begin{cases} (1, \ldots, 1 \mid 0, \ldots, 0 \mid 0, \ldots, 0) & \text{if } \tau = 0, \\ (\underbrace{0, \ldots, 0}_{7} \mid \underbrace{\tau, \ldots, \tau}_{7} \mid \underbrace{0, \ldots, 0}_{486}) & \text{otherwise,} \end{cases} \quad \lambda_j^{(\tau)} = 1000 \cdot \frac{e^{\theta_j^{(\tau)}}}{\sum_{j'=1}^{500} e^{\theta_{j'}^{(\tau)}}}. \quad (19)$$

The first block of 7 independent words are indicative of label 0, the second block of 7 *correlated* words are indicative of label 1, and the remaining 486 words are indicative of neither.

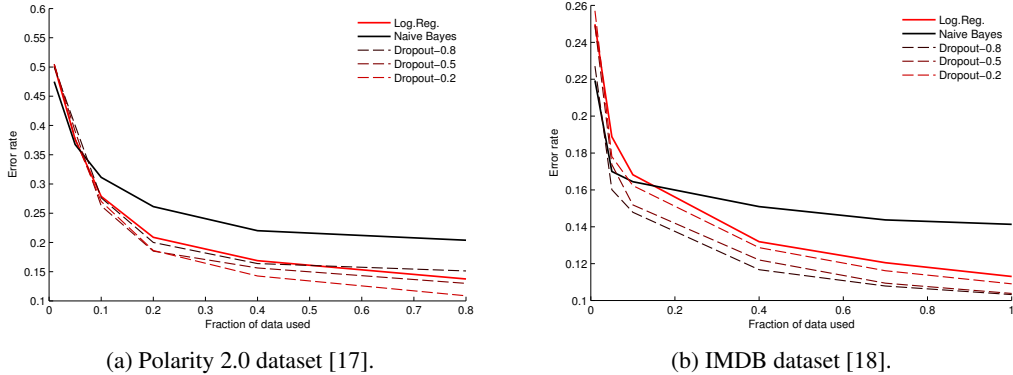

(a) Polarity 2.0 dataset [17].  (b) IMDB dataset [18].

Figure 3: Experiments on sentiment classification. More dropout is better relative to logistic regression for small datasets and gradually worsens with more training data.

We train a model on training sets of various size $n$, and evaluate the resulting classifiers' error rates on a large test set. For dropout, we recalibrate the intercept on the training set. Figure 2b shows the results. There is a clear bias-variance tradeoff, with logistic regression ($\delta = 0$) and naive Bayes ($\delta = 1$) on the two ends of the spectrum. For moderate values of $n$, dropout improves performance, with $\delta = 0.95$ (resulting in roughly 50-word documents) appearing nearly optimal for this example.

**Sentiment Classification**   We also examined the performance of dropout as a function of training set size on a document classification task. Figure 3a shows results on the Polarity 2.0 task [17], where the goal is to classify positive versus negative movie reviews on IMDB. We divided the dataset into a training set of size 1,200 and a test set of size 800, and trained a bag-of-words logistic regression model with 50,922 features. This example exhibits the same behavior as our simulation. Using a larger $\delta$ results in a classifier that converges faster at first, but then plateaus. We also ran experiments on a larger IMDB dataset [18] with training and test sets of size 25,000 each and approximately 300,000 features. As Figure 3b shows, the results are similar, although the training set is not large enough for the learning curves to cross. When using the full training set, all but three pairwise comparisons in Figure 3 are statistically significant ($p < 0.05$ for McNemar's test).

**Dropout and Generative Modeling**   Naive Bayes and empirical risk minimization represent two divergent approaches to the classification problem. ERM is guaranteed to find the best model as $n \to \infty$ but can have suboptimal generalization error when $n$ is not large relative to $d$. Conversely, naive Bayes has very low generalization error, but suffers from asymptotic bias. In this paper, we showed that dropout behaves as a link between ERM and naive Bayes, and can sometimes achieve a more favorable bias-variance tradeoff. By training on randomly generated sub-documents rather than on whole documents, dropout implicitly codifies a generative assumption about the data, namely that excerpts from a long document should have the same label as the original document (Proposition 4).

Logistic regression with dropout appears to have an intriguing connection to the naive Bayes SVM [NBSVM, 19], which is a way of using naive Bayes generative assumptions to strengthen an SVM. In a recent survey of bag-of-words classifiers for document classification, NBSVM and dropout often obtain state-of-the-art accuracies [e.g., 7]. This suggests that a good way to learn linear models for document classification is to use discriminative models that borrow strength from an approximate generative assumption to cut their generalization error. Our analysis presents an interesting contrast to other work that directly combine generative and discriminative modeling by optimizing a hybrid likelihood [20, 21, 22, 23, 24, 25]. Our approach is more guarded in that we only let the generative assumption speak through pseudo-examples.

**Conclusion**   We have presented a theoretical analysis that explains how dropout training can be very helpful under a Poisson topic model assumption. Specifically, by making training examples artificially difficult, dropout improves the exponent in the generalization bound for ERM. We believe that this work is just the first step in understanding the benefits of training with artificially corrupted features, and we hope the tools we have developed can be extended to analyze other training schemes under weaker data-generating assumptions.

## Footnotes

[1]Dropout training is known to work well in practice for multi-class problems [8]. For simplicity, however, we will restrict our theoretical analysis to a two-class setup.

[2] In topic modeling, the vertices of the simplex $\Theta$ are "topics" and $\tau$ is a mixture of topics, whereas we call $\tau$ itself a topic.

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
