[Supplementary Material]

# A Technical Results

We now give detailed proofs of the theorems in the paper.

## A.1 Altitude Training Phenomeon

We begin with a proof of our main generalization bound result, namely Theorem 1. The proof is built on top of the following Berry-Esseen type result.

**Lemma 5.** *Let $Z_1, ..., Z_d$ be independent Poisson random variables with means $\lambda_j \in \mathbb{R}_+$, and let*

$$S = \sum_{j=1}^{d} w_j Z_j, \ \mu = \mathbb{E}\left[S\right], \ and \ \sigma^2 = \text{Var}\left[S\right]$$

*for some fixed set of weights $\{w_j\}_{j=1}^{d}$. Then, writing $F_S$ for the distribution function of $S$ and $\Phi$ for the standard Gaussian distribution,*

$$\sup_{x \in \mathbb{R}} \left| F_S(x) - \Phi\left(\frac{x - \mu}{\sigma}\right) \right| \leq C_{BE} \sqrt{\frac{\max_j \{w_j^2\}}{\sum_{j=1}^{d} \lambda_j w_j^2}}, \tag{20}$$

*where $C_{BE} \leq 4$.*

*Proof.* Our first step is to write $S$ as a sum of bounded *i.i.d.* random variables. Let $N = \sum_{j=1}^{d} Z_j$. Conditional on $N$, the $Z_j$ are distributed as a multinomial with parameters $\pi_j = \lambda_j/\lambda$ where $\lambda = \sum_{j=1}^{d} \lambda_j$. Thus,

$$\mathcal{L}\left(S \mid N\right) \overset{d}{=} \mathcal{L}\left(\sum_{k=1}^{N} W_k \mid N\right),$$

where $W_k \in \{w_1, ..., w_d\}$ is a single multinomial draw from the available weights with probability parameters $\mathbb{P}\left[W_k = w_j\right] = \pi_j$. This implies that,

$$S \overset{d}{=} \sum_{k=1}^{N} W_k,$$

where $N$ itself is a Poisson random variable with mean $\lambda$.

We also know that a Poisson random variable can be written as a limiting mixture of many rare Bernoulli trials:

$$B^{(m)} \Rightarrow N, \ \text{with} \ B^{(m)} = \text{Binom}\left(m, \frac{\lambda}{m}\right).$$

The upshot is that

$$S^{(m)} \Rightarrow S, \ \text{with} \ S^{(m)} = \sum_{k=1}^{m} W_k I_k, \tag{21}$$

where the $W_k$ are as before, and the $I_k$ are independent Bernoulli draws with parameter $\lambda/m$. Because $S^{(m)}$ converges to $S$ in distribution, it suffices to show that (20) holds for large enough $m$. The moments of $S^{(m)}$ are correct in finite samples: $\mathbb{E}\left[S^{(m)}\right] = \mu$ and $\text{Var}\left[S^{(m)}\right] = \sigma^2$ for all $m$.

The key ingredient in establishing (20) is the Berry-Esseen inequality [see, e.g., 26], which in our case implies that

$$\sup_{x \in \mathbb{R}} \left| F_{S^{(m)}}(x) - \Phi\left(\frac{x - \mu}{\sigma}\right) \right| \leq \frac{\rho_m}{2s_m^3 \sqrt{m}},$$

where

$$s_m^2 = \text{Var}\left[W_k I_k\right],$$
$$\rho_m = \mathbb{E}\left[\left|W_k I_k - \mathbb{E}\left[W_k I_k\right]\right|^3\right],$$

We can show that

$$s_m^2 = \mathbb{E}\left[(W_k I_k)^2\right] - \mathbb{E}\left[W_k I_k\right]^2 = \frac{\lambda}{m}\mathbb{E}\left[W_k^2\right] - \left(\frac{\lambda}{m}\mathbb{E}\left[W_k\right]\right)^2, \text{ and}$$

$$\rho_m \leq 8\left(\mathbb{E}\left[|W_k I_k|^3\right] + \mathbb{E}\left[|W_k I_k|\right]^3\right) = 8\left(\frac{\lambda}{m}\mathbb{E}\left[|W_k|^3\right] + \left(\frac{\lambda}{m}\mathbb{E}\left[|W_k|\right]\right)^3\right).$$

Taking $m$ to $\infty$, this implies that

$$\sup_{x\in\mathbb{R}}\left|F_S(x) - \Phi\left(\frac{x-\mu}{\sigma}\right)\right| \leq \frac{4\mathbb{E}\left[|W|^3\right]}{\mathbb{E}\left[W^2\right]^{3/2}}\frac{1}{\sqrt{\lambda}}.$$

Thus, to establish (20), it only remains to bound $\mathbb{E}\left[|W|^3\right]/\mathbb{E}\left[W^2\right]^{3/2}$. Notice that $P_j \stackrel{\text{def}}{=} \pi_j w_j^2/\mathbb{E}\left[W^2\right]$ defines a probability distribution on $\{1,\dots,d\}$, and

$$\frac{\mathbb{E}\left[|W|^3\right]}{\mathbb{E}\left[W^2\right]} = \mathbb{E}_P\left[|W|\right] \leq \max_j\{|w_j|\}.$$

Thus,

$$\frac{\mathbb{E}\left[|W|^3\right]}{\mathbb{E}\left[W^2\right]^{3/2}} \leq \sqrt{\frac{\max_j\{w_j^2\}}{\sum_{j=1}^d \pi_j w_j^2}}.$$

$\square$

We are now ready to prove our main result.

*Proof of Theorem 1.* The classifier $h$ is a linear classifier of the form

$$h(x) = \mathbb{I}\{S > 0\} \text{ where } S \stackrel{\text{def}}{=} \sum_{j=1}^d w_j x_j,$$

where by assumption $x_j \sim \text{Poisson}\left(\lambda_j^{(\tau)}\right)$. Our model was fit by dropout, so during training we only get to work with $\tilde{x}$ instead of $x$, where

$$\tilde{x}_j \sim \text{Binom}\left(x_j,\, 1-\delta\right), \text{ and so unconditionally}$$

$$\tilde{x}_j \sim \text{Poisson}\left((1-\delta)\,\lambda_j^{(\tau)}\right).$$

Without loss of generality, suppose that $c_\tau = 1$, so that we can write the error rate $\varepsilon_\tau$ during dropout as

$$\varepsilon_\tau = \mathbb{P}\left[\widetilde{S} < 0 \,\middle|\, \tau\right], \text{ where } \widetilde{S} = \sum_{j=1}^d w_j \tilde{x}_j. \tag{22}$$

In order to prove our result, we need to translate the information about $\widetilde{S}$ into information about $S$.

The key to the proof is to show that the sums $S$ and $\widetilde{S}$ have nearly Gaussian distributions. Let

$$\mu = \sum_{j=1}^d \lambda_j^{(\tau)} w_j \text{ and } \sigma^2 = \sum_{j=1}^d \lambda_j^{(\tau)} w_j^2$$

be the mean and variance of $S$. After dropout,

$$\mathbb{E}\left[\widetilde{S}\right] = (1-\delta)\,\mu \text{ and } \text{Var}\left[\widetilde{S}\right] = (1-\delta)\,\sigma^2.$$

Writing $F_S$ and $F_{\widetilde{S}}$ for the distributions of $S$ and $\widetilde{S}$, we see from Lemma 5 that

$$\sup_{x \in \mathbb{R}} \left| F_S(x) - \Phi\left(\frac{x - \mu}{\sigma}\right) \right| \le C_{\mathrm{BE}} \sqrt{\Psi_\tau} \text{ and}$$

$$\sup_{x \in \mathbb{R}} \left| F_{\widetilde{S}}(x) - \Phi\left(\frac{x - (1 - \delta)\mu}{\sqrt{1 - \delta}\,\sigma}\right) \right| \le \frac{C_{\mathrm{BE}}}{\sqrt{1 - \delta}} \sqrt{\Psi_\tau},$$

where $\Psi_\tau$ is as defined in (9). Recall that our objective is to bound $\varepsilon_\tau = F_S(0)$ in terms of $\tilde{\varepsilon}_\tau = F_{\widetilde{S}}(0)$. The above result implies that

$$\varepsilon_\tau \le \Phi\left(-\frac{\mu}{\sigma}\right) + C_{\mathrm{BE}} \sqrt{\Psi_\tau}, \text{ and}$$

$$\Phi\left(-\sqrt{1 - \delta}\,\frac{\mu}{\sigma}\right) \le \tilde{\varepsilon}_\tau + \frac{C_{\mathrm{BE}}}{\sqrt{1 - \delta}} \sqrt{\Psi_\tau}.$$

Now, writing $t = \sqrt{1 - \delta}\,\mu/\sigma$, we can use the Gaussian tail inequalities

$$\frac{\tau}{\tau^2 + 1} < \sqrt{2\pi}\, e^{\frac{\tau^2}{2}}\, \Phi(-\tau) < \frac{1}{\tau} \text{ for all } \tau > 0 \tag{23}$$

to check that for all $t \ge 1$,

$$\Phi\left(-\frac{t}{\sqrt{1 - \delta}}\right) \le \frac{1}{\sqrt{2\pi}} \frac{\sqrt{1 - \delta}}{t} e^{-\frac{t^2}{2(1 - \delta)}}$$

$$= \frac{\sqrt{1 - \delta}\, t^{\frac{\delta}{1 - \delta}}}{\sqrt{2\pi}^{-\frac{\delta}{1 - \delta}}} \left(\frac{1}{\sqrt{2\pi}} \frac{1}{t} e^{-\frac{t^2}{2}}\right)^{\frac{1}{1 - \delta}}$$

$$\le 2^{\frac{1}{1 - \delta}} \frac{\sqrt{1 - \delta}\, t^{\frac{\delta}{1 - \delta}}}{\sqrt{2\pi}^{-\frac{\delta}{1 - \delta}}} \left(\frac{1}{\sqrt{2\pi}} \frac{t}{t^2 + 1} e^{-\frac{t^2}{2}}\right)^{\frac{1}{1 - \delta}}$$

$$\le \frac{2^{\frac{1}{1 - \delta}} \sqrt{1 - \delta}}{\sqrt{2\pi}^{-\frac{\delta}{1 - \delta}}} t^{\frac{\delta}{1 - \delta}}\, \Phi(-t)^{\frac{1}{1 - \delta}}$$

and so noting that in $t\,\Phi(-t)$ is monotone decreasing in our range of interest and that $t \le \sqrt{-2\log\Phi(-t)}$, we conclude that for all $\tilde{\varepsilon}_\tau + C_{\mathrm{BE}}/\sqrt{1 - \delta}\,\sqrt{\Psi_\tau} \le \Phi(-1)$,

$$\varepsilon_\tau \le \frac{2^{\frac{1}{1 - \delta}} \sqrt{1 - \delta}}{\sqrt{4\pi}^{-\frac{\delta}{1 - \delta}}} \left(\sqrt{-\log\left(\tilde{\varepsilon} + \frac{C_{\mathrm{BE}}}{\sqrt{1 - \delta}} \sqrt{\Psi_\tau}\right)}\right)^{\frac{\delta}{1 - \delta}}$$

$$\cdot \left(\tilde{\varepsilon} + \frac{C_{\mathrm{BE}}}{\sqrt{1 - \delta}} \sqrt{\Psi_\tau}\right)^{\frac{1}{1 - \delta}} + C_{\mathrm{BE}} \sqrt{\Psi_\tau}. \tag{24}$$

We can also write the above expression in more condensed form:

$$\mathbb{P}\left[\mathbb{I}\{\widehat{w} \cdot x^{(i)}\} \ne c_\tau \,\big|\, \tau^{(i)} = \tau\right] \tag{25}$$

$$= \mathcal{O}\left(\left(\tilde{\varepsilon}_\tau + \sqrt{\frac{\max\{w_j^2\}}{\sum_{j=1}^d \lambda_j^{(\tau)} w_j^2}}^{(1 - \delta)}\right)^{\frac{1}{1 - \delta}} \cdot \max\left\{1, \sqrt{-\log(\tilde{\varepsilon}_\tau)}^{\frac{\delta}{1 - \delta}}\right\}\right).$$

The desired conclusion (9) is equivalent to the above expression, except it uses notation that hides the log factors. $\square$

*Proof of Theorem 2.* We can write the dropout error rate as

$$\mathrm{Err}_\delta\left(\hat{h}_\delta\right) = \mathrm{Err}_{\min} + \Delta,$$

where $\text{Err}_{\min}$ is the minimal possible error from assumption (14) and $\Delta$ is the the excess error

$$\Delta = \sum_{\tau=1}^{T} \mathbb{P}\left[\tau\right] \tilde{\varepsilon}_\tau \cdot \left| \mathbb{P}\left[y^{(i)} = 1 \,\middle|\, \tau^{(i)} = \tau\right] - \mathbb{P}\left[y^{(i)} = 0 \,\middle|\, \tau^{(i)} = \tau\right] \right|.$$

Here, $\mathbb{P}\left[\tau\right]$ is the probability of observing a document with topic $\tau$ and $\tilde{\varepsilon}_\tau$ is as in Theorem 1. The equality follows by noting that, for each topic $\tau$, the excess error rate is given by the rate at which we make sub-optimal guesses, i.e., $\tilde{\varepsilon}_\tau$, times the excess probability that we make a classification error given that we made a sub-optimal guess, i.e., $\left| \mathbb{P}\left[y^{(i)} = 1 \,\middle|\, \tau^{(i)} = \tau\right] - \mathbb{P}\left[y^{(i)} = 0 \,\middle|\, \tau^{(i)} = \tau\right] \right|$.

Now, thanks to (14), we know that

$$\text{Err}_\delta\left(h_\delta^*\right) = \text{Err}_{\min} + \mathcal{O}\left(\frac{1}{\sqrt{\lambda}}\right),$$

and so the generalization error $\tilde{\eta}$ under the dropout measure satisfies

$$\Delta = \tilde{\eta} + \mathcal{O}\left(\frac{1}{\sqrt{\lambda}}\right).$$

Using (12), we see that

$$\tilde{\varepsilon}_\tau \leq \Delta \big/ \left(2\,\alpha\,p_{\min}\right)$$

for each $\tau$, and so

$$\tilde{\varepsilon}_\tau = \mathcal{O}\left(\tilde{\eta} + \frac{1}{\sqrt{\lambda}}\right)$$

uniformly in $\tau$. Thus, given the bound (11), we conclude using (25) that

$$\varepsilon_\tau = \mathcal{O}\left(\left(\tilde{\eta} + \lambda^{-\frac{1-\delta}{2}}\right)^{\frac{1}{1-\delta}} \max\left\{1, \sqrt{-\log\left(\tilde{\eta}\right)}^{\frac{\delta}{1-\delta}}\right\}\right)$$

for each topic $\tau$, and so

$$\eta = \text{Err}\left(\hat{h}_\delta\right) - \text{Err}\left(h_\delta^*\right) \tag{26}$$

$$= \mathcal{O}\left(\left(\tilde{\eta} + \lambda^{-\frac{1-\delta}{2}}\right)^{\frac{1}{1-\delta}} \max\left\{1, \sqrt{-\log\left(\tilde{\eta}\right)}^{\frac{\delta}{1-\delta}}\right\}\right),$$

which directly implies (16). Note $\eta$ will in general be larger than the $\varepsilon_\tau$, because guessing the optimal label $c_\tau$ is not guaranteed to lead to a correct classification decision (unless each topic is pure, i.e., only represents one class). Here, substracting the optimal error $\text{Err}\left(h_\delta^*\right)$ allows us to compensate for this effect. $\square$

*Proof of Corollary 3.* Here, we prove the more precise bound

$$\text{Err}\left(\hat{h}_\delta\right) - \text{Err}\left(h_\delta^*\right) = \mathcal{O}_P\left(\sqrt{\left(\frac{d}{n} + \frac{1}{\lambda^{(1-\delta)}}\right) \max\left\{1, \log\left(\frac{n}{d}\right)\right\}^{1+\delta}}^{\frac{1}{1-\delta}}\right). \tag{27}$$

To do this, we only need to show that

$$\text{Err}_\delta\left(\hat{h}_\delta\right) - \text{Err}_\delta\left(h_\delta^*\right) = \mathcal{O}_P\left(\sqrt{\frac{d}{n} \max\left\{1, \log\left(\frac{n}{d}\right)\right\}}\right), \tag{28}$$

i.e., that dropout generalizes at the usual rate with respect to the dropout measure. Then, by applying (26) from the proof of Theorem 2, we immediately conclude that $\hat{h}_\delta$ converges at the rate given in (17) under the data-generating measure.

Let $\widehat{\text{Err}}_\delta(h)$ be the average training loss for a classifier $h$. The empirical loss is unbiased, i.e.,

$$\mathbb{E}\left[\widehat{\text{Err}}_\delta(h)\right] = \text{Err}_\delta(h).$$

Given this unbiasedness condition, standard methods for establishing rates as in (28) [e.g., 27] only require that the loss due to any single training example $(x^{(i)}, y^{(i)})$ is bounded, and that the training examples are independent; these conditions are needed for an application of Hoeffding's inequality. Both of these conditions hold here. $\square$

## A.2 Distinct Topics Assumption

**Proposition 6.** *Let the generative model from Section 2 hold, and define*

$$\pi^{(\tau)} = \lambda^{(\tau)} / \left\| \lambda^{(\tau)} \right\|_1$$

*for the topic-wise word probability vectors and*

$$\Pi = (\pi^{(1)}, \dots, \pi^{(T)}) \in \mathbb{R}^{d \times T}$$

*for the induced matrix. Suppose that $\Pi$ has rank $T$, and that the minimum singular value of $\Pi$ (in absolute value) is bounded below by*

$$|\sigma_{\min}(\Pi)| \geq \sqrt{\frac{T}{(1-\delta)\lambda}} \left( 1 + \sqrt{\log_+ \frac{\lambda}{2\pi}} \right), \tag{29}$$

*where $\log_+$ is the positive part of $\log$. Then (14) holds.*

*Proof.* Our proof has two parts. We begin by showing that, given (29), there is a vector $w$ with $\|w\|_2 \leq 1$ such that

$$\mathbb{I}\left\{ w \cdot \pi^{(\tau)} > 0 \right\} = c_\tau, \quad \text{and} \quad \left| w \cdot \pi^{(\tau)} \right| \geq -\frac{1}{\sqrt{(1-\delta)\lambda}} \Phi^{-1}\left( \frac{1}{\sqrt{\lambda}} \right) \tag{30}$$

for all topics $\tau$; in other words, the topic centers can be separated with a large margin. After that, we show that (30) implies (14).

We can re-write the condition (30) as

$$\min\left\{ \|w\|_2 : c_\tau w \cdot \pi^{(\tau)} \geq 1 \text{ for all } \tau \right\} \leq \left( -\frac{1}{\sqrt{(1-\delta)\lambda}} \Phi^{-1}\left( \frac{1}{\sqrt{\lambda}} \right) \right)^{-1},$$

or equivalently that

$$\min\left\{ \|w\|_2 : S \Pi^\top w \geq 1 \right\} \leq \left( -\frac{1}{\sqrt{(1-\delta)\lambda}} \Phi^{-1}\left( \frac{1}{\sqrt{\lambda}} \right) \right)^{-1}$$

where $S = \mathrm{diag}(c_\tau)$ is a diagonal matrix of class signs. Now, assuming that $\mathrm{rank}(\Pi) \geq T$, we can verify that

$$\min\left\{ \|w\|_2 : S \Pi^\top w \geq 1 \right\} = \min\left\{ \sqrt{z^\top (\Pi^\top S^2 \Pi)^{-1} z} : z \geq 1 \right\}$$

$$\leq \sqrt{1^\top (\Pi^\top \Pi)^{-1} 1}$$

$$\leq |\sigma_{\min}(\Pi)|^{-1} \sqrt{T}$$

$$\leq \left( \frac{1}{\sqrt{(1-\delta)\lambda}} \left( 1 + \sqrt{\log_+ \frac{\lambda}{2\pi}} \right) \right)^{-1},$$

where the last line followed by hypothesis. Now, by (23)

$$\Phi\left( -\left( 1 + \sqrt{\log_+ \frac{\lambda}{2\pi}} \right) \right) \leq \frac{1}{\sqrt{2\pi}} \exp\left( -\frac{1}{2} \log \frac{\lambda}{2\pi} \right) = \frac{1}{\sqrt{\lambda}}.$$

Because $\Phi^{-1}$ is monotone increasing, this implies that

$$\left( 1 + \sqrt{\log_+ \frac{\lambda}{2\pi}} \right)^{-1} \leq \left( -\Phi^{-1}\left( \frac{1}{\sqrt{\lambda}} \right) \right)^{-1},$$

and so (30) holds.

Now, taking (30) as given, it suffices to check that the sub-optimal prediction rate is $\mathcal{O}\left(1/\sqrt{\lambda}\right)$ uniformly for each $\tau$. Focusing now on a single topic $\tau$, suppose without loss of generality that $c_\tau = 1$. We thus need to show that

$$\mathbb{P}\left[w \cdot \tilde{x} \leq 0\right] = \mathcal{O}\left(\frac{1}{\sqrt{\lambda}}\right),$$

where $\tilde{x}$ is a feature vector thinned by dropout. By Lemma 5 together with (11), we know that

$$\mathbb{P}\left[w \cdot \tilde{x} \leq 0\right] \leq \Phi\left(-\frac{\mathbb{E}\left[w \cdot \tilde{x}\right]}{\sqrt{\mathrm{Var}\left[w \cdot \tilde{x}\right]}}\right) + \mathcal{O}\left(\frac{1}{\sqrt{\lambda}}\right).$$

By hypothesis,

$$\mathbb{E}\left[w \cdot \tilde{x}\right] \geq -\sqrt{(1-\delta)\lambda^{(\tau)}}\Phi^{-1}\left(\frac{1}{\sqrt{\lambda}}\right),$$

and we can check that

$$\mathrm{Var}\left[w \cdot \tilde{x}\right] = (1-\delta)\sum_{j=1}^{d} w_j^2 \lambda_j^{(\tau)} \leq (1-\delta)\lambda^{(\tau)}$$

because $\|w\|_2 \leq 1$. Thus,

$$\Phi\left(-\frac{\mathbb{E}\left[w \cdot \tilde{x}\right]}{\sqrt{\mathrm{Var}\left[w \cdot \tilde{x}\right]}}\right) \leq \Phi\left(\Phi^{-1}\left(\frac{1}{\sqrt{\lambda}}\right)\right) = \frac{1}{\sqrt{\lambda}},$$

and (14) holds. $\qquad\square$

## A.3 Dropout Preserves the Bayes Decision Boundary

*Proof of Proposition 4.* Another way to view our topic model is as follows. For each topic $\tau$, define a distribution over words $\pi^{(\tau)} \in \Delta^{d-1}$: $\pi^{(\tau)} \stackrel{\text{def}}{=} \lambda^{(\tau)}/\|\lambda^{(\tau)}\|_1$. The generative model is equivalent to first drawing the length of the document and then drawing the words from a multinomial:

$$L_i \sim \mathrm{Poisson}\left(\|\lambda^{(\tau)}\|_1\right), \text{ and } x^{(i)} \mid \tau^{(i)}, L_i \sim \mathrm{Multinom}\left(\pi^{(\tau^{(i)})}, L_i\right). \qquad (31)$$

Now, write the multinomial probability mass function (31) as

$$\mathbb{P}_m\left[x; \pi, L\right] = \frac{L!}{x_1! \cdots x_p!} \pi_1^{x_1} \cdots \pi_d^{x_d}$$

For each label $c$, define $\Pi_c$ to be the distribution over the probability vectors induced by the distribution over topics. Note that we could have an infinite number of topics. By Bayes rule,

$$\mathbb{P}\left[x = v \mid y = c\right] = \mathbb{P}\left[L = \sum_{j=1}^{d} v_j\right] \cdot \int \mathbb{P}_m\left[v; \pi, \sum_{j=1}^{d} v_j\right] d\Pi_c(\pi), \text{ and}$$

$$\mathbb{P}\left[y = c \mid x = v\right] = \frac{\mathbb{P}\left[c\right] \int \mathbb{P}_m\left[v; \pi, \sum_{j=1}^{d} v_j\right] d\Pi_c(\pi)}{\sum_{c'} \mathbb{P}\left[c'\right] \int \mathbb{P}_m\left[v; \pi, \sum_{j=1}^{d} v_j\right] d\Pi_{c'}(\pi)}.$$

The key part is that the distribution of $L$ doesn't depend on $c$, so that when we condition on $x = v$, it cancels. As for the joint distribution of $(\tilde{x}, y)$, note that, given $\pi$ and $\tilde{L} = \sum_{j=1}^{d} \tilde{x}_j$, $\tilde{x}$ is conditionally $\mathrm{Multinom}(\pi, \tilde{L})$. So then

$$\mathbb{P}\left[\tilde{x} = v \mid y = c\right] = \mathbb{P}\left[\tilde{L} = \sum_{j=1}^{d} v_j\right] \cdot \int \mathbb{P}_m\left[v; \pi, \sum_{j=1}^{d} v_j\right] d\Pi_c(\pi), \text{ and}$$

$$\mathbb{P}\left[y = c \mid \tilde{x} = v\right] = \frac{\mathbb{P}\left[c\right] \int \mathbb{P}_m\left[v; \pi, \sum_{j=1}^{d} v_j\right] d\Pi_c(\pi)}{\sum_{c'} \mathbb{P}\left[c'\right] \int \mathbb{P}_m\left[v; \pi, \sum_{j=1}^{d} v_j\right] d\Pi_{c'}(\pi)}.$$

In both cases, $L$ and $\tilde{L}$ don't depend on the topic, and when we condition on $x$ and $\tilde{x}$, we get the same distribution over $y$. $\qquad\square$