[Reviews · NeurIPS 2014]

Submitted by Assigned_Reviewer_30

Dropout in deep neural networks is perturbing the output layer after non-linear mapping. Later it has been shown that this is closely related to perturbing the weight matrix before non-linear mapping due to the choice of mapping functions in neural networks (known as DropConnect). In a way, both Dropout and DropConnect perturb the decision boundary (ie parameter) after training, which is different from (though might be related to) perturbing the input feature vectors before training. This paper provides generalization bounds for training with perturbed features, which is more closely related to learning by feature deletion [13] or learning with corrupted features [9].

The noise considered in McAllester’s pac-bayes bound for dropout is a combination of Bernoulli step and Gaussian additive noise step, which is different from typical dropout setting, which uses Bernoulli multiplicative noise. The noise considered in this paper is different from all above, which may explain the improved bounds.
Despite the paper does not study under the same setting in previous work, the results are beneficial to the field to advance the understanding of dropout.
Summary: This paper improves the generalisation bound for dropout or more precisely learning with corrupted features using a different setting.

Submitted by Assigned_Reviewer_41

-------------------------------------------
review adjustment based on authors feedback
-------------------------------------------
In their response to the original review, the authors corrected my mistake and indeed the derivation is correct. Therefore, they have removed my main concern about this work.

There are some issues that are left unclear and I hope the authors have a chance to clarify them. In (1) for example, the behavior of the RHS as a function of \delta is unclear. In their response, the authors said that this formula is correct only for \delta=1/2. Therefore, the discussion about the bias-variance tradeoff cannot be driven by this formula.

------------------
Original comments
------------------
Dropout seems to be an important component of the recent success of deep neural nets. This study attempt to shed some light on the reasons for this success. This line of research is of significance if it will allow us to come up with better regularization techniques. The authors study dropouts in the context of topic modeling where the data is assumed to be generated by a specific process. The main contribution of this work is a theoretical analysis that suggest that dropouts should increase accuracy.

The paper is well written and contains novel contributions. The topic is relevant to the NIPS community. Unfortunately, there is a main component of the analysis that I fail to understand. In page 4, line 204 it is claimed that the variance after dropouts is (1-\delta) times the original variance (the same argument is repeated in page 11 line 593 of the appendix). However, IMHO, the variance is (1-\delta)^2 times the original variance. This is so since if X and Y are random variables such that Y= bX where b is a scalar than the Var(Y)=b^2 Var(x). Am I wrong about it? Is it possible to fix the proof despite this issue?

Another way to see this problem is looking at Figure 1b. It is implied from this figure that if X~N(\mu,\sigma^2) then the probability P[X < 0] is smaller than the probability P[Y < 0] where Y=(1-\delta)X. However, it is clear that these probabilities are identical as long as (1-\delta) > 0.

I think that the presentation can be improved in (1) and (9). Following these formulas, in line 63 and in line 192 it says "the constants in the bound depend only on \delta". If a constant depend on \delta it is probably not a constant. This presentation requires the discussion in line 64 that explains why the bound does not behave as you may expect when \delta is close to one. Instead, making the dependency on \delta more explicit may be useful. One way to do it, that will keep the formula readable is to add a constant \Kappa to the R.H.S. in (1) and (9) and define \Kappa in a separate equation.

In line 88 it is claimed that delta allows control over the bias-variance tradeoff. However, in line 81 it is claimed that the dropout does not introduce much of a bias. These claims seem to contradict each other. It would be helpful to clarify this issue.

The theoretical analysis studies the zero-one loss but the experiments use the logistic loss (line 117). It would be nice to have some arguments to support these choices.

In line 170 there seems to be a typo in the text: "the error on the original measure is actually much smaller than the error on the original measure"

In (7) and (8) it is not clear if this is the training error or test error.

In section 6 (Experiments and Discussion) there is no statistical significance analysis of the results. The plots don't contain error bars and therefore it is hard to tell if the results are artifact of randomness or do they represent a real phenomenon.

Summary: The paper studies dropouts in the context of linear classifiers in the text domain with focus on theory. This is an important topic since we are still looking for good explanations for the success of dropouts.

Submitted by Assigned_Reviewer_44

This paper gives a generalization bound for dropout training of document classifiers based on thresholding linear functions of empirical word frequencies. The theorem shows that under simple assumptions about the data distribution that the excess risk under dropout classification is smaller than the excess risk of ERM.

While the observations are clear and interesting,I have a variety of issues that I hope the authors can address in their feedback.

1) The paper ignores regularization. Baldi et al. and Wager et al. both make the simple observation that for linear regression dropout training is equivalent to ridge regression with the strength of the regularization is governed by the dropout rate. Admittedly, the current paper addresses classification error rate rather than L_2 loss. However, the regression case strongly indicates that h^*_delta is very different from h^* --- h^*_delta is effectively regularized while h^* is not. If dropout is a form of regularization then the main effect is in the difference between the generalization error rates of h^*_delta and h^*. This "main effect" is not addressed at all in this paper.

2) The paper suggests that the dropout analysis improves generalization bounds from the VC bound of tilde{O}(sqrt{d/n} to tilde{O}{d/n}. I think this is misleading due to the failure to properly treat regulirazation. A proper treatment of regularization (see McAllester's tutorial) yields excess risk bounds bounds that go as O(C(h)/n) where C(h) is the complexity of classifier h (such as sparsity).

3) The paper suggests that any generalization bound on excess risk can be "pumped" by the main theorem. So a linear bound on excess risk --- such as the O(1/n) bounds of McAllester --- should yield bounds that go as O(1/n^2). But of course this is not true. The issue is again the proper treatment of regularization.

4) From informal the proof of theorem 1 it is clear that the generalization error of h is closely related to the margin (as in many standard generalization bounds). But the margin is then completely ignored in the analysis. Again, the fundamental issue seems to be that the analysis ignores regularization and ignores the fundamental difference difference between h^*_delta and h^*.
Summary: The technical content of this paper is a clear. However, I think that the discussion is misleading and the technical result does not actually provide much insight. A similar comment probably applies to McAllester's bound as his result also primarily concerns the excess risk.
Author Feedback
Author rebuttal: Thank you for your helpful and encouraging reviews. Since each reviewer commented on fairly different aspects of the paper, we separate our response into 3 parts.

-----------------------------------------

R1: Thank you for your valuable perspective on our paper. It is exciting to see how many different forms of dropout have been recently found to work well.

In our experience, doing binomial dropout (as in [9, 13]) vs Bernoulli dropout appears to behave very similarly in practice. The reason we study binomial dropout is that it pairs nicely with our Poisson generative model for word counts. We believe, however, that the high-level insights from our paper should hold for other forms of dropout too.

-----------------------------------------

R2: The main concern is about the variance of dropout predictions used during training. As explained below, we want to reassure the reviewers that our expression for the variance of dropout predictions is not a mistake, and that the expressions on line 204 of our manuscript are correct. We appreciate your close reading of our paper, and for giving us a chance to respond to your concerns.

During training, dropout does not involve multiplicatively down-weighting the training features: \tilde{Xi} is not equal to (1 - \delta) Xi. Rather, as described in equation (3) of the paper, dropout down-samples features binomially: \tilde{Xi} ~ Binom(Xi, 1 - \delta). The extra noise from binomial sampling adds some variance; this is why the variance of \tilde{Xi} is (1 - \delta) Xi rather than (1 - \delta)^2 Xi.

We can check the variance formula formally as follows. By the Binomial variance formula, E[Var[\tilde{Xi} | Xi]] = delta * (1 - delta) E[Xi], whereas Var[E[\tilde{Xi} | Xi]] = (1 - delta)^2 Var[Xi]. Now, in the Poisson model, E[Xi] = Var[Xi], and so by the law of total variance,
Var[\tilde{Xi}] = Var[E[\tilde{Xi} | Xi]] + E[Var[\tilde{Xi} | Xi]] = (1 - delta) Var[Xi]
By an extension of this argument, the variance of the dropout prediction w * \tilde{X} is (1 - delta) times the variance of the raw prediction w * X. We can add a remark following line 204 of our paper deriving the variance formula.

For the rest, we thank you for the suggestions regarding presentation!

- In (1) and (9), the results only hold for a fixed pre-defined choice of delta (e.g., delta = 0.5). We will emphasize this in our revision.

- What we meant is that dropout gives a "good" bias-variance trade-off, i.e., it lets us cut variance a lot without inducing too much bias. We can clarify.

- This choice of 0-1 loss for theory / logistic loss for experiments has been used by, e.g., Ng and Jordan in their paper on discriminative vs generative classifiers (NIPS, 2002); the relation between these two losses has been studied by Zhang (Ann. Statist., 2004) and Bartlett, Jordan, and McAuliffe (JASA, 2006). We will add references.

Finally, thank you for pointing out typos and spots where our text was not clear. In Figure 2, the statistical error is negligible (we averaged across many trials) but we will give a standard error estimate; in Figure 3, we will discuss statistical significance.

-----------------------------------------

R3: We are grateful for your thought-provoking comments! We had also grappled with many of these issues while preparing our manuscript, and would be very interested to discuss them further.

(1) We were in fact originally trying to provide generalization bounds for dropout in the "regularization" framework, but didn't get any strong results. The beginning of the present paper was the insight that we can get strong results by studying the "document shortening" aspect of dropout instead of the regularization aspect. We think that reconciling these two points of view would be very interesting (and have spent many days trying to do so), but have not found a way to do so yet.

(2) We agree - this is very surprising. Classical generalization bounds are all of the form O(C(h)/n). But our analysis shows that, in the delta = 0.5 case, taking n training examples and chopping them randomly in half gives us a generalization bound that behaves as though we had n^2 examples! We are still looking for ways to get a deeper insight into this phenomenon.

(3) To get quadratic (1/n^2) convergence, we do not need to use PAC Bayesian bounds. In fact, if we use delta = 0.75, then Corollary 3 gives us (d/n)^2 bounds. Now, we want to emphasize that the bound we proved was O((d/n)^2 + 1/sqrt(lambda)), where lambda scales with document length. Without this extra lambda term, our result would impossible --- just as you say!

Our bound is most natural in a regime where the d/n term and the lambda term scale away at the same rate, i.e., documents get longer as lambda ~ (n/d)^(1/(1 - delta)). For example, if delta = 0.5 and d ~ n^{4/5}, then we would want lambda ~ n^{2/5}, i.e., we would want to have documents of length roughly lambda ~ sqrt{d}. We will add discussions like this to the manuscript.

As to McAllester's PAC-Bayesian bounds - it is not immediately clear to us whether we can used our result to pump them up, as the bounds deal with distributions over hypotheses rather than point hypotheses. The key technical step would involve showing a PAC-Bayesian dropout analogue to (28) in the appendix. This would be an interesting topic for further work.

(4) We agree - our analysis is closely related to margin. The assumption that codifies this is our "distinct topics" Assumption 3. We should add the word "margin" to the discussion of that assumption.

Again, thank you for your very constructive comments. We apologize for our rather short responses due to the limitations of the present medium. We hope to have many chances for conversations along these lines while presenting our paper.